# Knowledge Mapping and Institutional Prospects on Circular Carbon Economy Based on Scientometric Analysis

**DOI:** 10.3390/ijerph191912508

**Published:** 2022-09-30

**Authors:** Zhengai Dong, Lichen Zhang, Houjian Li, Yanhui Gong, Yue Jiang, Qiumei Peng

**Affiliations:** 1School of Law, Chongqing University, Chongqing 400044, China; 2College of Economics, Sichuan Agricultural University, Chengdu 611130, China

**Keywords:** circular carbon economy, scientometric analysis, institutional prospects

## Abstract

The circular carbon economy is receiving increasing research attention as an essential tool for reducing carbon emissions and mitigating climate change. However, there is no research on the literature distribution and the current situation of the circular carbon economy studies. This paper presents a scientometric analysis of 1452 academic papers on the circular carbon economy and their references from 2010–2021 using the Citespace visualization network. The results show that research on the circular carbon economy has experienced a relatively gradual growth from 2010 to 2016, followed by an explosive growth from 2016 to 2021. Research cooperation among countries is close, forming a relatively concentrated cooperation network, while the core author group has not yet formed. Furthermore, the research on circular carbon economy strongly correlates with relevant international hotspots and national policy changes, reflecting the instrumental characteristics of circular carbon economy research. We summarized three main research topics through keywords clustering. In addition, we point out the future research directions from technical progress considering industry differences and cooperation, multiple environmental policies and legal system construction, interregional and international cooperation, etc., from an institutional research perspective. This article provides an essential and valuable reference for related research.

## 1. Introduction

Circular Carbon Economy (CCE) is a comprehensive system that relies on managing and cutting carbon dioxide emissions through an integrated management system of the environment and economy with the support of modern technology [1]. Greenhouse gas emission reduction has gradually become the focus of academic circles facing the intensification of global climate problems [2,3]. In 1992, the United Nations Framework Convention on Climate Change first put forward the common but differentiated responsibilities in carbon emission reduction, which contained the meaning of cutting greenhouse gas. The Kyoto Protocol restricted greenhouse gas emissions as a rule for the first time in 1997, and more than 170 countries ratified it [4]. The Paris Agreement officially proposed carbon neutrality in 2015, requiring member countries to actively reduce emissions to achieve the long-term goal of climate change within 2 °C and the global temperature rise limit of 1.5 °C. [5] As of January 2022, the total number of countries that have declared their commitment to net zero emissions or carbon neutrality has reached 136 after the COP26 of the United Nations Climate Change Conference in 2021. In addition, 24 major cities and 683 major companies announced their commitment to achieving net zero emissions or carbon neutrality [6]. However, the control of total carbon emissions is not optimistic globally. The British Petroleum Company (BP) statistics disclosed that global carbon emissions had continued to rise since 2000, reaching a new high of 34.36 billion tons by 2019 [7]. The global carbon emissions dropped in 2020 under the influence of COVID-19, which decreased to 32.28 billion tons, down 5.8% year-on-year, but began to rebound at the end of the year [8]. According to the 2021 Global Climate Report at the 26th United Nations Climate Conference, the carbon dioxide concentration reached a new record globally in 2020, and the situation was still deteriorating in 2021 [9]. It is self-evident that human activities have an impact on future climate trends. To control global warming caused by artificial factors at a specific level and achieve the goal of net zero carbon emissions, it is necessary to limit the cumulative carbon dioxide and other greenhouse gas emissions [10]. Closed-loop carbon management is one of the most essential means for achieving net zero carbon emissions [11].

Circular carbon economy provides an ideal scheme by storing, utilizing, and removing carbon dioxide. In 2016, economist McDonough William innovatively put forward “circular carbon economy”, which refers to all activities involving CO_2_ emission reduction, reuse, recycling, and removal [12,13,14]. Traditional consumption follows a linear production, consumption and emission pattern, and is maintained unsustainably [15]. Circular economy provides the best output in the form of protecting existing resources by converting end-of-life goods into other resources. Its advantage is that waste production and resource use are minimized, thus having higher economic and environmental benefits [16]. However, in the recycling process, there are also ecological risks that cause secondary pollution and the waste of resources. The sustainable principle makes resource efficiency a key strategy in achieving economic prosperity and combat climate change [17]. The circular economy is an economic growth model characterized by recycling and resource conservation. Based on the principle of achieving resource reduction, recyclability and reuse, circular economy meets the requirements of sustainable development and can significantly improve resource efficiency. It aims to realize a new closed-loop economic model of “exploitation-production-waste reuse-production”, which makes material and energy utilized reasonably and continuously in the process of the economic cycle. Circular carbon economy considers both carbon productivity and resource output rate, and its essence is the recycling and reduction in carbon. Based on the circular economy, circular carbon economy has become a new trend to cope with climate change [18]. For example, the European Union (EU) set the 2050 climate-neutrality target in the first European Climate Law on March 2020 [19,20]. Studies have also shown that countries developing circular economies will reduce greenhouse gas emissions by 70% [21]. As the earliest country to promote circular carbon economy, Saudi Arabia has achieved tremendous energy and emission reductions and will continue to seek more cost-effective options [22]. In 2021, the G20 Protection and Climate Stewardship Working Group conference aimed to substantially balance the benefits of carbon emissions by circular carbon economy.

Developing circular carbon economy can not only reduce carbon emissions per unit product [23,24] but also alleviate the shortage of resources [25], which is beneficial to the realization of carbon emission reduction targets among countries and regions in the world. It also provides a perfect path to respond to global climate change and build a circular development economic and policy system [26]. There is a growing interest globally in circular carbon economy to work towards the goal of peak CO_2_ emissions and carbon neutrality. The circular carbon economy was put forward and developed under the background of climate warming, so the research mostly focused on practical applications, including technology and management. In terms of technology, industrial research and development have been intensified, aiming to transfer various industrial sectors in the past few years [12], especially in manufacturing [27] and mining [28]. The latest research on technology includes plastic recycling [29], waste gasification and pyrolysis [30], metal ore refining [31], chemical recycling [32], solid recycling, etc. [33]. The research on the technical route of circular carbon economy is focused on the carbon capture [34] and converting greenhouse gases into useful products, such as carbon-negative chemical production from gas fermentation [35]. In terms of management, it is mainly promoted from the circular chain and policies, such as supply chain management [36], life cycle [37], various models to assess the future energy path and its impact on climate change [38], national performance [39], institutional cooperation [40], human factors [41], policy evolution [42], role [43] and improvement, etc. [44].

However, no research systematically describes the overall research content of circular carbon economy, such as the development track, knowledge base, applicable fields, and future development space. The current research fails to fully respond to the practical difficulties and needs. In recent years, there has been a surge in research on circular carbon economy, with a focus on the alternative energy, utilization efficiency and carbon recovery of heavily polluting industries such as plastics and agriculture. Due to the limitations of empirical technology or the barriers of different industries, most studies are restricted to a certain section or a certain source, lacking an overall grasp and rational thinking of the further development the of circular carbon economy, which affects the depth and expansion of related studies. In particular, what are the influential or mainstream technologies and opinions in the field of circular carbon economy? Is the carbon reduction effect as expected? How can we preserve the development achievements of circular carbon economy through policies and laws? It can be seen that the goal of achieving zero emissions has not been matched with the relevant research and policies, and even the systematic reviews and summaries of the existing research are lacking. Therefore, based on reviewing all the existing research, we provide a direction for formulating policies to achieve these goals. This article tried to sort out the knowledge base and cooperation status, clarify the hot research frontier and development trend of circular carbon economy, evaluate the prominent, problems and put forward suggestions for the future development of circular carbon economy. Through bibliometric analysis of the literature on circular carbon economy, this paper presents the time, country, institution, author, keywords and other information involved in this research field in a visual form. We further make a detailed interpretation of the information and analyze the evolution of the core and hot issues and the potential research direction in the future to provide a reference for policymakers and relevant researchers.

## 2. Materials and Methods

### 2.1. Research Methods

Pritchard, a British information scientist, proposed bibliometrics as a research method that uses bibliographic information as the research object and bibliography as the theoretical foundation [45]. By quantitative analysis of literature information, we can statistically summarize and grasp the development trend, the research focus and the overall situation of an academic topic. Advances in research methods and interdisciplinary developments have facilitated the expansion of bibliometric analysis in revealing the structure of knowledge and exploring cutting-edge trends in a particular discipline.

CiteSpace (version 5.8 R3) is a citation visualization analysis software developed by Professor Chen Chaomei from the School of Information Science and Technology, Drexel University, Philadelphia, PA, USA [46]. Its core function can be summarized as: with the help of a visualization map of the evolution of the knowledge field, it can reflect and approach the scientific development law of a specific field in the physical world more profoundly in a more abstract category of “second-order subject” through more vivid and intuitive images. It is not only helpful in explaining the existing scientific discoveries, but also conducive to the establishment of new scientific discoveries based on literature [47]. Specifically, Citespace presents the structure, rules and distribution of research content by visual means, which helps in finding the hidden patterns and rules of knowledge structure in specific disciplines and fields. It also leads the researchers to study research frontier and knowledge in related fields by observing the research situation related to this field visually [48]. As a new bibliometric analysis tool to explore the subject knowledge structure and frontier trend, it is widely used in the literature review. However, in a large network, the relationships between nodes are complex, which will affect the visualization effect.

### 2.2. Data Sources

The circular carbon economy has evolved from the circular economy and low-carbon circular economy. If it is searched as its title, research deviation and data loss are hard to avoid, which makes it challenging to cover all related research in the field of circular carbon economy. Therefore, we search on the research topic of circular carbon economy, including title, abstract, author keywords and keyword plus. The data source is the web of science (WOS) database, and the search time is up to 31 December 2021. In the WOS database, we set “TS = (“circular carbon economy ”),” source category = core collection (SCI, SSCI only) without setting the time range. A total of 1452 available documents on circular carbon economy from 2010–2021 were finally obtained after data screening and manual pretreatment. The literature in 2022 and non-academic papers were excluded such as press releases and other documents inconsistent with the research topic. The effective literature is exported in the plain text format. The literature on circular carbon economy is analyzed through cluster analysis, co-occurrence analysis, catastrophe value analysis, etc., to present cooperation status, research hotspots and research frontiers. Among the available literature, the literature on circular carbon economy is mainly about “circular economy” and “low-carbon circular economy”, as well as its coupling study.

### 2.3. Parameter Explanation

We use CiteSpace to process the literature, take six years as a time slice, then set the top 300 levels of most cited items and select the top 10% of the most cited items from each time slice. Set the C, CC, and CCV thresholds to (1, 1, 20), (3, 3, 20), and (3, 3, 20), respectively. C (Citation) means the lowest count of citations. Only the literature that satisfies this requirement is allowed to proceed below. CC (Co-citation) is the co-citation times, and CCV (co-citation cosine coefficient) is the co-cited times after standardization. After running, the author collaboration map is obtained, and the size of nodes in the map is directly proportional to the author’s articles. Then, we import the literature mentioned above into Citespace software, take one year as a time slice, then set the top 50 items and select the top 10% as the most cited in each time slice at the same time, as well as setting the thresholds of C, CC, and CCV to (2, 3, 15), (3, 3, 20), and (3, 3, 20). After running, we obtain the organization cooperation map, national or regional cooperation map, literature co-citation map, keyword co-occurrence map, keyword clustering map, etc.

“Co-occurrence” refers to the phenomenon that the information of a document appears together. The characteristic items include the external and internal characteristics, such as title, author, keywords, organization, etc. Before the visual analysis, the Citespace parameter is shown as follows. Frequency is the number of times that knowledge units appear in the literature data. Higher frequency usually means more significant influence. Burst refers to the heat status of a single knowledge unit. The sharp increase or decline in a certain year indicates its sudden change, which reflects that the research on this piece of knowledge is a hot or unpopular subject. It has been used to identify sudden changes in events and other types of information. Degree centrality represents the number of connections between nodes. The stronger the centrality is, the closer the node is to others. Betweenness centrality is defined as the number of times a node serves as the closest bridge between two other nodes, whose height is inversely proportional to the importance of nodes. Nodes with high betweenness centrality tend to identify the boundary crossing potential that may lead to transformative discovery [49]. In addition, N indicates the number of network nodes, E indicates the number of connections, and the network density objectively indicates the close connection between nodes. The sigma score of a node is a combined measure of the betweenness centrality and burst of the node (i.e., the cited reference) in CiteSpace. The higher the centrality and prominence is, the higher the sigma value of the node is. Most of the sigma values are 1.00, indicating that it is very important in structure and citation change. According to the clarity of network structure and clustering, modularity statistics and silhouette statistics can be used to judge the mapping effect. Generally speaking, the Q value is generally in the range of [0, 1], and Q > 0.3 indicates that the community structure is significant. The S value of is between [1, −1]. The higher the silhouette score, the more consistent the cluster members are. When the value of S is 0.7, clustering is efficient and convincing. If it is above 0.5, clustering is generally considered reasonable [50]. In order to make the picture beautiful and readable, we selected the ideal atlas as the final result according to modularity statistics and silhouette statistics.

### 2.4. The Research Route

For analyzing convenience, the exported data contains complete literature date, including author, title, publications, abstract, keywords and cited references. Figure 1 explains Citespace’s analysis process, main functions and our research prospects based on the above analysis. According to the logical framework shown in Figure 1, this paper analyzes the authors, institutions, countries and regions, references and keywords, respectively, and obtains the general research situation, cooperation situation, research hotspots and trends, etc., and further puts forward institutional prospects based on the data analysis.

## 3. Results and Discussion

### 3.1. Basic Feature Analysis

To a certain extent, the number of published documents can reflect the development speed and research process of related research on circular carbon economy. Figure 2 shows the distribution of academic papers on circular carbon economy (since 2022 is the year of the search ends, the publications are shown until December 2021). Overall, the total publications on circular carbon economy are trending upwards, from 1 in 2010 to 724 in 2021. Before 2016, the research on circular carbon economy was generally stable. With the increasingly serious climate problem, circular carbon economy has become the focus of more and more countries and scholars as one of the important ways to decrease greenhouse gas emissions. The development of circular carbon economy has become more and more critical with the increasing attention to carbon emission reduction, so the research on circular carbon economy has been moving at a rapid pace, indicating that the academic interest in this field is growing.

In addition, as can be seen from Table 1, the related research is primarily concentrated in the field of environmental science with prominent technical characteristics, involving many disciplines such as materials, energy, biotechnology, chemistry, physics, etc. It also shows that technological progress and innovation are indispensable and basic supports for the realization of circular carbon economy.

### 3.2. Author’s Cooperation Map

The author’s cooperation map reflects the number of papers and cooperation of scholars in the field of circular carbon economy and can identify authors with high activity and strong influence in this field. Different nodes represent different authors. The bigger the node is, the more articles are published. The connection of nodes represents the collaboration among authors. Zabaniotou Anastasia, Hou HongYing, Bialowiec Andezej and other scholars can be clearly identified in Figure 3 and Table 2. In other words, they are the most prolific authors in this field. Among them, Zabaniotou Anastasia has published the most documents, mainly on the bio-economic direction based on waste, such as evaluating the potential role of bio-economy [51] and advocating the value of food waste in circular bio-economy [52]. There are a large number of Chinese scholars, accounting for two-thirds in Table 3. A research group represented by Hou HongYing has formed, and its cooperation with Chinese scholars is higher, while its cooperation with international authors with high-quality research results is lower. China has a large base of researchers, which is one of the reasons for the remarkable number of Chinese authors. Moreover, the carbon neutrality and the peak carbon dioxide emissions commitment announced by China’s government at the United Nations General Assembly triggered the upsurge of Chinese scholars’ research on circular carbon economy which may also be a crucial reason. It can be seen that the collaboration among authors is mostly concentrated in 2016–2021, and there are relatively few connections between nodes. Citespace only considers the co-citation of the first author when calculating the co-citation, and the same author is cited once in the same document, so the network density is low. Lack of cooperation platform, cooperation motivation and geographical constraints lead to little international cooperation. In the case of cooperation, there is less research by a single author, and it is more common for scholars to cooperate. The degree of cooperation between scholars with high publications and other scholars is also closer, and a fixed cooperation circle has been formed. Scholars’ cooperation can improve research efficiency, which is one of the root causes of the obvious correlation between scholars’ cooperation and output quantity.

The concept of “core group of authors” from Price’s Law can judge whether a specific discipline has formed a stable core group of authors to present a cooperative research state of authors [53]. Price’s Law states that “half of the papers (50%) are written by a group of high-productivity people, and the number of core authors is equal to the square root of all authors”. Therefore, the authors with more than three high-quality papers on circular carbon economy in the recent 12 years are the core authors. By analyzing 1452 literature, the total number of core authors who meet the requirements is 208. The core authors have published 796 papers, accounting for 54.8%, which shows that the core team has been constituted and there is close collaboration among authors.

### 3.3. Institutional Cooperation Map

Table 3 shows that the League of European Research Universities has the most literature, with four of the top ten being university research institutions, but the participation of social institutions is low. Many researchers in colleges and universities are usually the main force of academic research on some topics that lack funds and concerns of enterprises. Each node in Figure 4 means a research institution, and the connection between nodes represents the academic cooperation between institutions. Figure 4 shows that the research on circular carbon economy shows “overall dispersion and local aggregation”. The network density is relatively low, meaning the frequency of inter-institutional cooperation is low. The research network is “dispersed” as the normal state due to different regions and research emphases. Some institutions have conducted more research in related fields and exerted great influence. Research institutions, such as the Chinese Academy of Sciences, Delft University, CSIC, and Aristotle University Thessaloniki, etc., have become “local aggregation” points, which relate to the cooperation of neighboring institutions. For example, the institutional cooperation between Chinese Academy of Sciences, Imperial College London and Shanghai Jiaotong University is an essential factor that promote its achievements.

As can be seen from Figure 4, research institutions at the edge have a low degree of cooperation and few articles, and most of them come from well-known foreign universities, such as Leeds University, Lisbon University and Milan Polytechnic University. Universities with solid strength have also displayed decisive leadership in research on circular carbon economy. In the tunnel effect theory, the environmental Kuznets curve presents the dynamic link between GDP expansion and environmental degradation as an inverted “U” curve [54,55]. “Environmental pollution” is an inevitable negative externality in economic development, and the natural environment in developed countries is obviously better. Such marginal institutions are usually located in developed countries or regions, where the environment, economy and resource exploitation have been decoupled. The development of circular carbon economy is relatively mature, which may explain why most marginal institutions are in developed countries. The figure also shows that Chinese research institutions are at the core of this field and play a special role in connecting other institutions. The existing literature shows that China’s research is mainly concentrated in environmental science, accounting for 51.26%, mainly taking regional industries as the research object, and proposing the cost-saving path of energy and resources [56]. Some papers summarized the path of China’s low-carbon economy, including circular economy, low-carbon cities, energy industry, trading market, reduction targets, afforestation and rebound, low-carbon technologies, etc. [57]. The statistics of BP Company indicate that China is currently the largest carbon emitter globally. The carbon emissions in the Asia-Pacific region accounted for 52% of the overall emissions world widely, of which China accounted for 30.7% in 2020 [7]. The shortage of resources per capita, the uneven distribution of resources and the current situation of carbon emission determine that China urgently needs to develop circular carbon economy and improve resource efficiency. Globally, organizations such as the International Energy Agency (IEA), the International Association of Energy Economics (IAEE) and the Organization of Petroleum Exporting Countries (OPEC) have taken an interest in circular carbon economy and have reported on the concept in detail, supporting it as a systematic approach to achieving economic growth, energy market stability and sustainable development in the post-epidemic era [58].

### 3.4. National or Regional Cooperation Mapping

Figure 5 shows the national or regional cooperation network map, with nodes representing different countries or regions. The node connection means a cooperative relationship between countries or regions. Nodes represent the analyzed objects, and the more frequently they appear, the larger the nodes will be. The color and thickness of the inner circle indicate the frequency of occurrence in different periods. The connection between nodes indicates the co-occurrence relationship, and its thickness indicates the intensity of co-occurrence. Then the color corresponds to the time when the node co-occurs for the first time. The change from the cool to the warm indicates the change in time from early to recent. This graph has been able to show the knowledge clusters, the relationships among clusters and their evolution over time. We can observe the countries or regions with more research on circular carbon economy and its time intuitively. At the same time, we can also observe countries or regions gradually conducting research on carbon circular economy. From the node size, aggregation degree and timeline, the research in the past 10 years has shown a parabolic stage feature of “emergence-outbreak-decline”. Due to the fact that the environmental problem has gradually become a global problem, compared with the cooperation between institutions and authors, the cooperation between countries is relatively close, so its network density is relatively higher.

From Table 4, the research is dominated by China and developed countries. Among them, China, Italy, UK, Spain and USA are top five with the highest number of articles. However, not only developed countries paid attention to circular carbon economy; rather, developing countries have gradually joined the research sequence. Some international treaties required developed countries to adopt the common but differentiated responsibility to offer financial, technical and scientific research assistance to developing countries, for achieving scientific and rapid carbon emission reduction [59]. This has led to a sudden research increase in developing countries after 2016, such as India and Malaysia. Brazil mainly focuses on environmental science, environmental engineering and green sustainable science and technology. For example, Caldas and Lucas Rosse study building materials, trying to find alternatives to cutting the construction industry’s carbon emissions [60]. India mainly explores environmental science, fossil energy and environmental engineering. For example, Agrawal, Rohit and others investigate how to reduce environmental pollution and break down barriers to reducing carbon emissions in the automobile industry [61]. Due to loose constraints, different countries have different responses to the topic of carbon emission reduction. Moreover, the basis and pace of developing circular carbon economy in countries with varying levels of development are not synchronized, which may explain the differences among countries.

Figure 5 also shows that European countries have studied this field earlier, such as Germany, Finland and Denmark. In 2010, the EU issued the “EU 2020 Strategy”, which takes improving resource utilization rate and developing circular economy as crucial strategies for sustainable development. Subsequently, the EU issued the “Circular Economy Package Plan” in 2015, which laid the cornerstone of the strategic concept of circular economy [62]. During this period, European countries made rich research achievements. Despite the high-level of economy and industry in Europe, many European countries are still seriously relying on fossil fuels [63]. It is estimated that European countries will retain only about 14% of proven oil, 72% of proven coal and 18% of proven natural gas reserves by 2050 [64]. Therefore, it is foreseeable that energy security and developing circular carbon economy will become significant global topics.

As seen in Figure 5 and Table 4, China has published the largest number of articles on circular carbon economy. Although China started late, it has obtained more research results and occupied the first place in quantity. China’s research on circular carbon economy strongly correlates with policy progress, such as the Cleaner Production Promotion Law in 2003, national plan to address climate change, which was the first pilot project in low-carbon provinces and cities in 2007, and the Circular Economy Promotion Law in 2009. Therefore, the early research focused on the significance, concept and connotation [65], ways and modes [66] of low-carbon economy development. After China’s formal accession to the Paris Agreement in 2016, the “Circular Development Leading Plan” and other policies, China’s research on circular carbon economy has exploded in order to respond and satisfy low-carbon practice, which has also promoted the application and migration of circular economy to circular carbon economy.

Figure 5 also shows that research cooperation among countries is close, forming a centralized cooperation network. The research priorities of various countries tend to be similar, including bio-energy, environmental remediation, nanotechnology, medicine and food, nano-carbon materials, mine tailings, etc., which is helpful for researchers to reach cooperation. For example, British scholar Ibn-Mohammed cooperated with Malaysian and Nigerian scholars to critically analyze the impact of coronavirus disease on global economy and ecosystem in 2019 and the opportunities for circular economy strategy [67]. Scholars from Germany and the United States, who were early movers in developing a circular carbon economy, study future low-carbon electricity systems by integrating life cycle assessment and comprehensive energy modelling. It is the general tendency to develop circular carbon economy and jointly cope with climate change through international cooperation [68].

### 3.5. Co-Citation Map and Analysis of Literature

Henry Small first put forward the concept of co-citation. Co-citation refers to two (or more) articles cited by one or more articles simultaneously, so these articles are called co-citation relationship [69]. All co-cited articles constitute the knowledge base, and the research frontier is composed of the collection of cited documents that cite these knowledge bases [70]. The literature’s citation frequency can reflect the research hotspots and trends of related disciplines. By observing literature co-cited data, we can more directly understand the frontier issues of this discipline.

Figure 6 and Figure 7 are the cluster view maps of literature co-citation. As a prominent member of the developing knowledge milestone and professional knowledge base, the cluster has recurring topics in its articles to reflect the relationship between the knowledge base and the research frontier. On the one hand, a research frontier may remain in the same co-citation group; on the other hand, the research frontier may belong to different majors and become the knowledge base of new majors. As far as the contour score is concerned, the homogeneity of the largest cluster is slightly lower than that of the smaller cluster. From 0 to 7, the smaller the number is, the more keywords are included in the cluster. Each cluster is composed of several closely related words. When two or more co-cited documents are cited in the same article, they are connected. Therefore, the links between co-cited documents are relatively few, the network density is relatively low, and academic exchanges need to be strengthened [69]. In Figure 6, the connection between nodes represents the co-citation relationship, and the thickness of the connection represents the co-citation strength between them. The thicker the connection is, the stronger the co-citation relationship will be. The red node is a mutation node, which indicates the value of the cited quantity of the node that had a significant change in the corresponding period. Figure 7 shows the keyword clustering of co-cited articles and their distribution. The main clustering keywords are Guiyang China, circular economy, plastic waste, to-energy system, etc. The smaller the tag serial number, the more keywords the cluster contains, so “Guiyang China” contains the most keywords. Guiyang was chosen as the first pilot city for circular economy in 2005, put forward the “Eco-economic City”, and formulated the “Master Plan of Eco-economic City Construction in Guiyang (2006–2020)”. Then, Guiyang was listed as a national pilot city for ecological civilization construction in 2009, the Guiyang Conference on Ecological Civilization was held in August of the same year, and the Guiyang Consensus was published, which attracted the attention of domestic and foreign researchers [71]. Circular economy strategy is generally considered an effective way to realize low-carbon transformation of cities. The practice of obtaining low-carbon benefits through circular economy in Guiyang is an essential practice of low-carbon transformation, which has significant reference value.

Figure 8 is the timeline map of co-cited literature on circular carbon economy. The abscissa is the corresponding year, and the right side is the cluster label of co-cited documents. The timeline map can intuitively see the first citation time of an article containing the corresponding keyword. The radius of the circle represents the accumulation of the citation frequency of the article. The concept of circular carbon economy is an organic extension of circular economy and is consistent with the objectives of low-carbon economy. Circular economy attaches importance to the circulation of material, energy and water in economic flow, and circular carbon economy further focuses on energy and carbon flow. Then, the articles with the keyword “circular economy” have the highest citation frequency, and experienced explosive growth after 2015, indicating that circular economy still has a dominant position in circular carbon economy. This may be because the development of circular carbon economy is still based on circular economy and still within the macro framework of circular economy.

According to the algorithm of Kleinberg’s burst-detection, if the citation frequency of a paper suddenly increases rapidly, then the safest explanation is that the paper has hit a pivotal point in the complex system of academic field [72]. Table 5 is the burst map of the co-citation literature, which reflects the top 15 pieces of the literature with the highest burst value, indicating that the article has changed violently in a certain period. Haas et al. (2015) burst from 2016 to 2020, which is co-cited for the longest duration. The paper tracked all social material flowed in the 27 countries after 2005 and evaluated the global resource cycle using a socio-metabolic method. As a result of the use of the new method, the article was taken seriously for a long time after publication [73]. The latest burst is Jambeck et al. (2015) [74], which focuses on marine plastic pollution. It points out that the cumulated plastic waste will harm the marine environment and affect the resource recycling efficiency. Marine environmental problems have caught the eye of the world in recent years, and its viewpoints cater to the research frontier, and get some attention after 2019.

Table 6 is the co-citation table derived from Citespace, which contains cited frequency, emergence value, degree centrality, betweenness centrality, author, publication year and other information of cited references. It can be seen from the table that there are three pieces of literature with high burst values.

Ghisellini et al.’s (2016) study received wide attention from 2017 to 2019, which reviewed the literature of the twenty years before its publication, and discussed the essential characteristics and prospect of circular economy, such as its origin, basic fundamentals, strengths and weaknesses, modes, implementation patterns in the world, which is a critical processing section of circular carbon economy after that [78]. Steffen et al. (2015) was published after the 45th Davos Forum in 2015 and has further updated the planetary boundaries framework based on the expert research and scientific progress in the five years prior to its publication. It introduces a two-tier approach to the definition of some boundaries and points out that no matter which boundary is broken, the Earth system will enter a new state. The extension of PB theory in this paper has been paid attention to in the field of ecological science, and is significant for the research of environmental science [79]. Pan et al. (2015) pointed out that the waste-to-energy chain provided a viable path to address the predicament of global energy requirements, rubbish disposal and carbon emission [80]. Based on the evaluation of several leading technologies, it put forward comprehensive policies and strategies for effectively establishing of the WTE supply chain and reviewed and explained several successful experiences of the WTE supply chain with detailed explanations. Since the publication of this article, the citation frequency has obviously increased, especially after 2017, which shows that the problems and countermeasures put forward in this article have been a continuous research hotspot in recent years and have high research value. From the time point of view, the impetus of international conferences and agreements has led to a shift in research on the circular economy towards environmental and sustainable development issues, which has developed rapidly since 2015. These documents will also be of great significance to the future development of circular carbon economy.

### 3.6. Analysis of Keywords and Hotspots

Keywords can reflect the key points, core issues, and the hotspots in specific research fields. The analysis object is DE and ID fields in the document, and the result is keyword co-occurrence network, which can reflect the current research hotspots in a certain field and what hotspots have been produced in the past. This part of the literature is mainly based on the results of keyword co-occurrence, showing the hot research direction. The distribution and progression of research hotspots gives a visual indication of the changes in research topics, perspectives and approaches over time. The keyword co-occurrence map takes one year as a time slice. The threshold is set to 3 by frequency, and the nodes are adjusted by clustering to accentuate the high-frequency keywords of circular carbon economy research. The final display effect is as follows (Figure 9).

Figure 9 controls the number of displayed nodes according to the keyword frequency. It can be observed that keywords such as “circular economy” and “reverse logistics” are the central nodes in Figure 9. The larger and darker center circle indicates that the research around the keywords such as circular economy, reverse logistics, anaerobic digestion, municipal solid waste recycling and carbon is more frequent and numerous. The high intermediary center of the nodes indicates that the above keywords play a stronger role in connecting the surrounding nodes through connecting lines and are important terms for connecting the surrounding keywords. At the same time, the literature retrieval time of this research started in 2010, so the keywords of circular carbon economy that appeared in 2010 cannot measure their burst value. As a result, the keywords in 2010 are not observed and are denoted as “-”. The high-frequency keywords are sorted out as shown in the following figure (Table 7).

As can be seen from Table 7, “circular economy”, “life cycle assessment”, “carbon”, “performance”, “energy”, and “management” are keywords frequently used in the research on circular carbon economy in the period. The keywords with the highest frequency include circular economy and carbon, which is consistent with the development logic of circular carbon economy, that is, circular carbon economy is based on circular economy with carbon as its object. Life cycle assessment is a method to evaluate a product or service’s environmental impact in its whole life cycle, which coexists with the concept of circular economy. Circular carbon economy plays a role in different stages of the lifecycle. This kind of keywords embodies the following characteristics: (1) “Circular economy”, “carbon” and “carbon dioxide” and other keywords related to circular carbon economy appeared in 2011, which indicates that the related study on the concept started earlier, and the research results are relatively large; (2) keywords related to circular carbon economy mainly appeared in 2010–2015, such as “carbon”, “emission”, “carbon footprint”, “carbon dioxide” and “activated carbon”, etc. In this period, the research direction mainly combined carbon-related indicators and circular economy. From 2015 to 2022, many scholars shifted their research direction to combining circular economy with other disciplines to achieve pollution reduction and carbon reduction. Their research directions were more detailed, such as environmental science, green sustainable technology, engineering environment, chemistry, biotechnology and other disciplines; (3) emerging dynamic concepts and underlying research questions often reflect the frontiers of related disciplines. As research hotspots, “Bioma” and “technology” have experienced a certain period of explosion. In addition, if the centrality of the node exceeds 0.1, it means that the node is the central node, which is important and has great influence in the research, including “performance”, “anaerobic digestion”, “municipal solid waste”, “emission”, “adsorption” and “environmental impact” in Table 7. It can be seen that there is more research on the above keywords, with greater influence. The centrality of high-frequency keywords is generally low, indicating that although they appear frequently, they do not have strong intermediary functions on other keywords. In order to further express the relationship between time and keywords, the timeline view is used to convert the above Figure 9 and Table 7, and the display effect is as follows (Figure 10).

The timeline view expresses the evolution of main keyword frequencies in different years and the connection between keywords. The node where keywords are located indicates the time and frequency. As can be seen from Figure 10, keywords represented by “emission” and “circular economy” appeared around 2011, with complex connections and relatively mature development. During the thirteen years of observation, such high-frequency keywords had little influence on the other keywords appearing later. In comparison, the keywords have been gradually progressive since then, with relatively stable development and small change. From the above figures and tables, it is evident that the research on circular carbon economy is gradually developing from the whole to the parts, from a single discipline to multi-disciplines. The research direction shifted from the construction of a macro-knowledge structure to the analysis of specific paths of sub-disciplines. At the same time, cross-disciplinary research and refined research are increasingly evident.

According to the above analysis of keywords, we generalize three important research directions:

First, concept evolution and connotation explanation. Gao et al. (2010) paid early attention to low-carbon economy through circular agricultural economy [95]. Williams (2019) proposed that circular carbon economy is beneficial to the balance of carbon cycle as a closed system, and further achieves the goal of mitigating climate change. The circular economy’s concept is one of the earliest research hits within the circular carbon economy [96]. Bonviu (2014), Stahel (2016) and Geissdoerfer et al. (2017) shifted the perspective of theoretical study from low-carbon economy to circular economy, concentrating on circular economy’s concept [15,21,81]. Kirchherr et al. (2017) critically discussed various concepts of circular economy, which put more emphasis on the economic level, followed by the environmental quality. They pointed out that the previous definitions hardly responded to social equity and intergenerational impact [83]. Murray et al. (2017) traced the economic and ecological origin and predecessor of circular economy, and discussed its performance in commerce and policy [92]. Then, the relationship between carbon cycle and sustainable development principle became a research hotspot. Ghisellini et al. (2016) clearly sorted out different types of relationships between “circular economy” and “sustainability” and analyzed their similarities and differences [78]. Korhonen et al. (2018) defined circular economy in the viewpoint of sustainable development. They pointed out that it could attract business groups and policy-making groups to participate in sustainable development work from the perspective of environmental sustainability, but it still faced challenges such as unclear concepts [82]. By discussing the definition and principle of circular economy, the focus has shifted from economic dimension to the environmental impact level, which provides much theoretical basis for circular carbon economy.

Second, multi-disciplinary accelerating measures of circular carbon economy. The related literature on circular carbon economy has apparent interdisciplinary characteristics, with more interactions with other disciplines. Under the theme of circular carbon economy, scholars mainly study industries such as food waste, agriculture, transportation and energy consumption [97]. In recent years, the number of interdisciplinary studies with carbon cycling as the research starting point or purpose has increased, which provides multidisciplinary technical and knowledge support for the realization of circular carbon economy. For example, Genovese et al. (2017) first suggested combining circular economy principle with continuous supply chain management [84]. Pomponi & Moncaster (2017) demonstrated the contribution of circular economy research’s theoretical basis on the building environments [90]. Meyer et al. (2018) and Lee et al. (2017) proposed developing a novel sustainable value chain by interlinking energy, chemistry and waste management, which holds out a feasible and forward-looking prospect for achieving net zero carbon [12,14]. There is also literature concerned with the efficiency of the means to realize the circular carbon economy, such as Alshammari (2020), Grim et al. (2020), Meys et al. (2021), Seidl et al. (2021). They have explored feasible paths for transforming linear carbon economy into circular carbon economy from different angles, such as reducing the lifecycle of plastics and using other materials to suppress greenhouse gas emissions. It reflects the research direction of circular carbon economy in recent years [2,24,98,99]. In addition, some scholars have summarized the practical experience of individual countries on circular carbon economy. For example, Mathews and Tan (2016), Su et al. (2013) and Geng et al. (2012) reviewed China’s practice. They pointed out that China has made outstanding contributions to promoting waste reuse to reduce carbon emissions by setting goals, adopting policies, financial measures and legislation especially in the past decade [77,100,101]. Lah (2016) introduced Germany’s policies on circular carbon economy [102]. McDowall et al. (2017) compared the differences between China and Europe in formulating relevant policies [103].

Third, improvement direction and development trend of circular carbon economy. The literature mainly focuses on management, system and continuity. Mansouri et al. (2020) emphasized that the realization of circular carbon economy requires joint efforts and cooperation among countries [104]. Alsarhan et al. (2021) proposed that the circular carbon economy may be achievable by high transparency of worldwide management, which supports advanced technologies that can effectively utilize energy [1]. Bonsu (2020) pointed out that the low-carbon closed-loop business model should develop legislation, cooperation, investment and motivation mechanisms to reach global targets [11]. Furthermore, many scholars have researched improving the support for circular carbon economy’s improvement. For example, Rizos et al. (2019) held that the EU needs a solid and transparent carbon accounting system as a catalyst for investment in low-carbon processes [105]. Geng et al. (2019) suggested an international platform to exchange information and experience, and to facilitate the collaboration of industrial policies [26]. Bakshi et al. (2021) emphasized the role of supervision and finance [106]. In addition, scholars have given their views on the future path to developing circular carbon economy. Lieder & Rashid (2016) discussed the development prospects from resource shortage, waste generation and economic advantages, and built a comprehensive circular economy framework for the renewable economy and natural environment and formulated a practical implementation strategy [89]. Kalmykova et al. (2018) analyzed different paths and basic principles of circular economy, focusing on two tools suitable for developing circular economy, namely circular economy strategy database and circular economy implementation database [91]. Korhonen et al. (2018) further put forward a research model of circular economy, which is used to limit the imbalance problems and enhance the contribution of circular economy to global sustainable development [82]. Moraga et al. (2019) proposed that the monitoring of single indicators should be developed into comprehensive indicators [94].

## 4. Conclusions and Institutional Prospects

### 4.1. Conclusions

The formation and improvement of the circular carbon economy concept have a specific policy background and research foundation. With the sustainable development strategy becoming the world’s trend, climate change makes the sustainable development of carbon face significant challenges. The circular economy has been integrated with environmental protection, clean production and green consumption as theoretical foundations of circular carbon economy. The literature on circular carbon economy is intimately linked to the change in global climate patterns, the academic research level of scholars, relevant national environmental policies and social hotspots. This study uses Citespace information visualization techniques to analyze 1452 literature and reveals the overall situation of circular carbon economy from 2010 to 2021 in SCI and SSCI of WOS, from the perspectives of literature quality, author groups, research institutions, research hotspot changes, literature co-citation and keywords. We concluded that the research on circular carbon economy experienced a relatively slow growth from 2010 to 2016, and an explosive growth from 2016 to 2021. The research direction also shifted from the construction of macro-knowledge structure to the analysis of micro-disciplinary content, and the characteristics of interdisciplinary and refined research became increasingly prominent. According to Kuhn’s scientific revolution structure, scientific progress is an iterative revolutionary process, which includes several stages: normal science, crisis and revolution. In the normal scientific stage, the research in a field is mainly dominated by a specific scientific paradigm, including a consensus on a set of theories, methods and research agendas. In the crisis stage, anomalies become inevitable, challenging the foundation of the current paradigm. Alternative and competitive paradigms are developed to solve exceptions. In the revolutionary stage, the new paradigm is mature enough to replace the existing paradigm which obviously can’t cope with the urgent crisis. It provides the overall framework for the research community [107,108]. The growth trend of circular economy responds to the new problems, demands and anomalies of climate change. Moreover, 2016 is a turning point in time, which may be related to the Paris Agreement in 2015 and the formal proposal of the concept of circular carbon economy in 2016. The popularity of scientific research on circular carbon economy is positively correlated with relevant policies, showing a certain policy orientation. Since then, countries have paid more and more attention to carbon emission reduction and circular carbon economy and have responded to the peak carbon dioxide emissions and carbon neutral targets. So, the research on carbon circular economy has experienced explosive growth. Based on the changing characteristics of the number of posts, it can be divided into two periods. The first period is a process from scratch from 2010 to 2016, during which the concept of circular carbon economy began to enter the researchers’ field of vision. The number of publications is relatively small, probably because the connotation of the concept of carbon cycle has not been unified. The second period is 2016–2021, with the number increasing year by year, and the number of published articles increasing from 24 to 663. This may be related to the growth of researchers in this field and policy orientation, but the decisive factor is the development of the circular carbon economy itself.

The number of authors’ articles and their cooperation reflected that the core group of authors of circular carbon economy research had not yet formed, and there was less cooperation among authors from various countries, whilst there was more cooperation among Chinese scholars. The participation of social institutions on circular carbon economy is low, and universities hold the mainstream positions of academic research on circular carbon economy. The matching degree between research theory and practice remains to be observed according to the analysis of institutions’ publications and cooperation. Through the analysis of the distribution and cooperation of countries or regions, it is concluded that the research priorities and primary coverage of various countries tend to be similar, and the research cooperation among countries in the world is close, forming a relatively robust cooperation network. Based on the literature co-citation and keyword analysis, we have revealed and summarized three critical topics and development paths including the concept evolution and connotation explanation, multi-disciplinary governance measures, improvement direction and development trend of circular carbon economy. The research reflects the influence of relevant international hotspots and national policy changes, significantly responds to social concerns in a specific period, and reflects the instrumental characteristics of circular carbon economy research.

This research still has some limitations. First of all, the data range of this study is limited, which needs to be further expanded. In this paper, the literature in SCI and SSCI of WOS core database is used as the data source, and it is difficult to avoid the statistical errors caused by the small database scope. In future research, a larger database can be included to ensure the comprehensiveness and accuracy of the data. Secondly, this study takes English as the main language, only analyzes journals in English, and lacks research on differences among countries. According to the research conclusion, some countries have made outstanding achievements in the research of circular carbon economy. Constrained by language barriers, conducting specific analysis for individual countries is difficult. Finally, this study focuses on hot keywords to study the trend of hotspots evolution, and the hotspots evolution at keyword level needs further study.

### 4.2. Institutional Prospects

As a critical means to reduce carbon emissions and effectively alleviate climate problems, circular carbon economy has attracted more and more attention from researchers, and related high-quality research results are gradually increasing. The present research situation of circular carbon economy contains multiple inspirations for future research:

First, research on the promotion path of circular carbon economy considering industrial differences and cooperation. A full-fledged circular carbon industry and scientific and technological innovation system can provide the technical support to realize the circular carbon economy and its ultimate realization. It is necessary to construct a specific and feasible path to realize carbon cycle in all phases of industry development, and to achieve effective connection and cooperation from the production end to the consumption end to form a sustainable carbon cycle development model. Different industries have significant differences in pollution emissions. However, scholars mostly focus on the research of circular carbon economy in heavily polluting industries, such as plastics and agriculture [109]. Not only are they short of in-depth discrimination of product supply and demand characteristics, carbon reduction efficiency, industrial science and technology level, and environmental protection foundation differences in different industries, but they also pay little attention to the efficiency improvement of digital technology application in different industries, and even neglect the research of differentiation paths and collaboration among industries at different stages and links in industrial chains or clusters currently. From the production side, carbon input can be reduced by using carbon-containing wastes as secondary carbon raw materials [32]. From the consumption side, the development and utilization of low-carbon energy or alternative energy sources are studied to reduce the use of carbon [110], and the research on ways to improve the efficiency of carbon capture and storage in different industries should be further strengthened. In addition, on the basis of evaluating the performance of varying waste management schemes in terms of carbon emission reduction, cost, environmental sustainability, consistency of methods, feasibility and reproducibility [111], we can formulate more efficient and targeted technical routes in line with industry characteristics, and turn theoretical deduction into practical progress to explore the carbon emission reduction potential of different industries. There is still much room for research on the more detailed path of the future circular carbon economy.

Second, research on multiple environmental policies and institutional construction of circular carbon economy. Since the practice and policy of circular carbon economy have been strengthened, theoretical research, especially considering some fundamental problems, has become insufficient. Since the specific path to achieve carbon neutrality by relying on circular economy has not yet been clear, there is a lack of theoretical basis for constructing and perfecting the medium-and long-term development strategic plan of circular carbon economy and its governance system for achieving carbon neutrality. On the one hand, it is necessary to clarify further the interactive relationship between environmental policies and the development of circular carbon economy. There is a lack of research on the role of circular carbon economy and the factors affecting the role of environmental policies. Especially under the long-term implementation, the research on the stability and practical effects of policies is insufficient. In the future, researchers should strengthen their understanding of environmental policy interaction, comprehensively promote the knowledge system construction of circular carbon economy and carbon neutrality and provide academic support for the scientific formulation of construction projects, plans and policies. On the other hand, the realization of circular carbon economy does not depend on gradual changes, but on the systematic upgrading of social and legal systems. Legal changes such as energy price reform have positive significance for carbon reduction [43]. The imperfect governance system of circular carbon economy hinders the implementation of carbon emission reduction measures and the realization of carbon cycle. In the future, it is requisite to enhance further the research on responsibility system, public service, management system, evaluation index systems, economic statistics, standard systems, resource consumption identification systems, etc. to support life cycle management. At the same time, it is also necessary to strengthen the research on ecological effects and positive functions of incentive policies, alleviate the ineffectiveness of administrative orders, and form a multi-interactive co-governance system of “government-market-social organizations” [112].

Third, research on interregional and international cooperation of circular carbon economy. The practice progress of international cooperation is accompanied by different degrees of research climax, which reflects the inspiring significance of interregional and international cooperation for the research of circular carbon economy, providing more perspectives and theoretical preparations for international and interregional cooperation and practice expansion in respect of circular carbon economy. At present, many countries have adopted different measures and means to reduce carbon emissions, such as Germany, France, Britain, New Zealand and other countries through legal provisions, Norway, Switzerland, Japan and other countries through policy announcements, and Costa Rica, Fiji, Uruguay and other countries through submitting independent emission reduction commitments to the United Nations. It has created a policy environment and institutional foundation for international cooperation in circular carbon economy. The International Carbon Action Partnership, the International Renewable Energy Agency and the United Nations have set up platforms for interregional and international cooperation, offering a certain political foundation for the multi-level and multi-channel national consultation mechanism. The goal of zero carbon promotes establishing partnerships in basic science and politics. However, in the circular carbon economy field, especially in improving operational economic policies and exchange mechanisms, there is still a lot of room for international and interregional cooperation in the future. One is the innovation of the cooperation form. Future national and regional cooperative research needs to continue to explore innovative ways, bring developed countries’ experience and capacity accumulation into full play, and provide ideas for expanding the advantages of circular carbon economy in some areas. We must explore new ways of international exchange and cooperation from the aspects of policy standard formulation, market function and technological innovation and application, and strengthen the cooperation and dialogue between global carbon market and carbon pricing, in order to promote the innovation of means and modes of the new international carbon cycle development order. The other is the transformation of cooperation results. The transformation path of extending laboratory research results to larger factories and the win-win cooperation through resource exchange are issues that require attention to in future research. Through the cooperative technology research and development and promotion of circular carbon economy, the link between the green circular industry and the international market will be energized.

## Figures and Tables

**Figure 1 ijerph-19-12508-f001:**
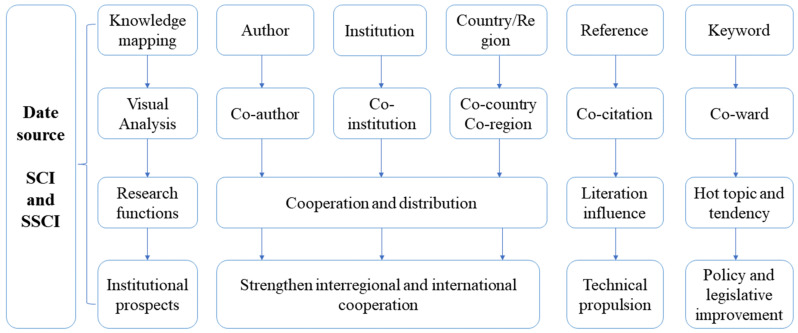
Research roadmap.

**Figure 2 ijerph-19-12508-f002:**
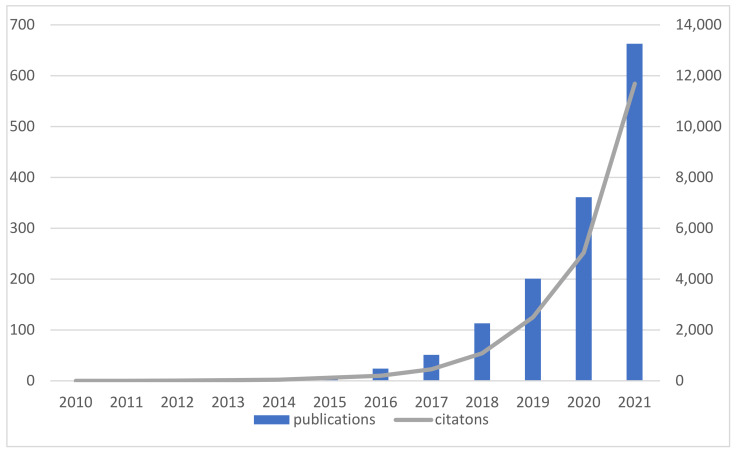
Annual distribution of the overall distribution trend of academic papers.

**Figure 3 ijerph-19-12508-f003:**
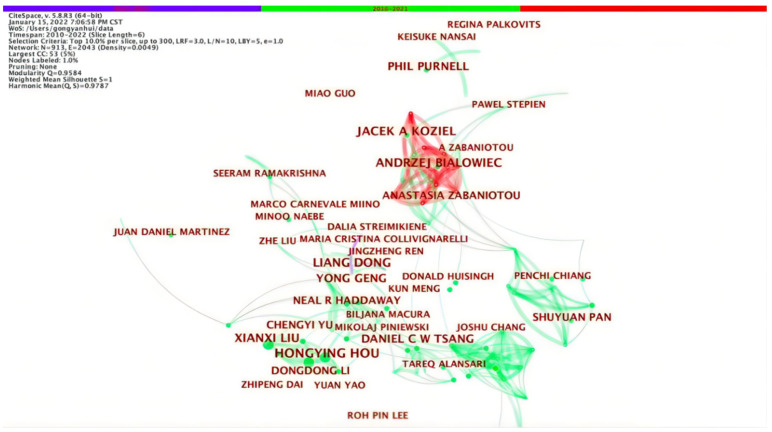
Author’s cooperation network.

**Figure 4 ijerph-19-12508-f004:**
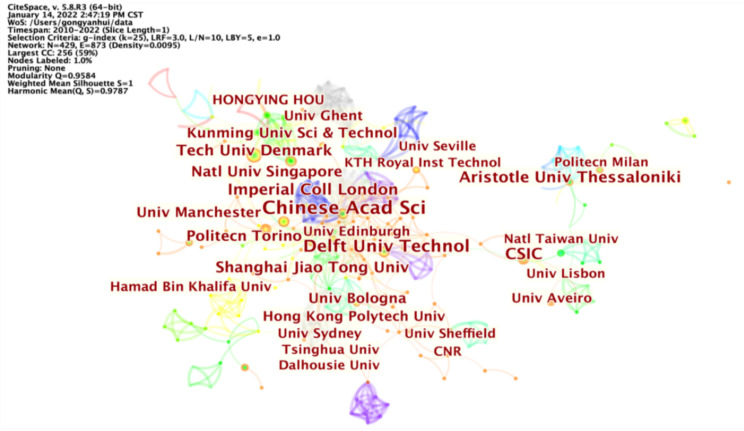
Institutional cooperation network.

**Figure 5 ijerph-19-12508-f005:**
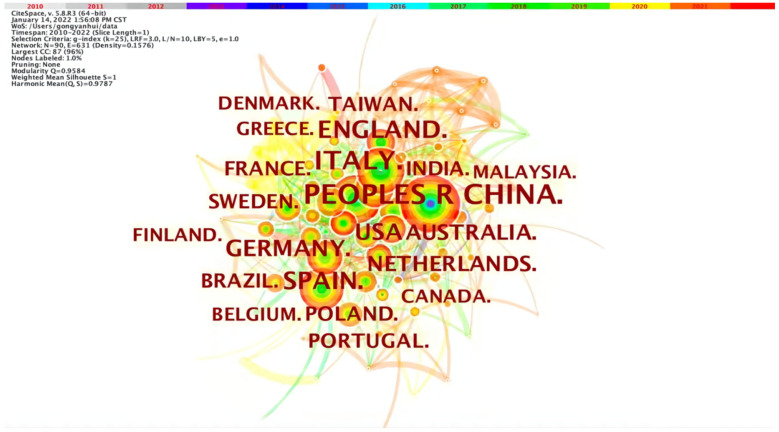
National or regional cooperation networks.

**Figure 6 ijerph-19-12508-f006:**
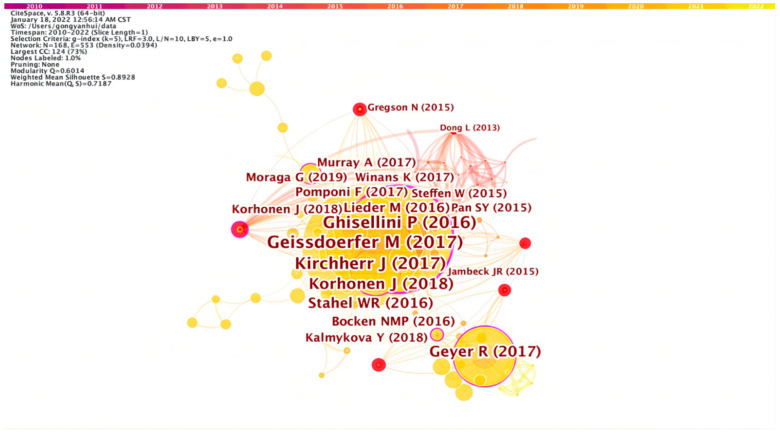
Co-citation literature map (2010–2021).

**Figure 7 ijerph-19-12508-f007:**
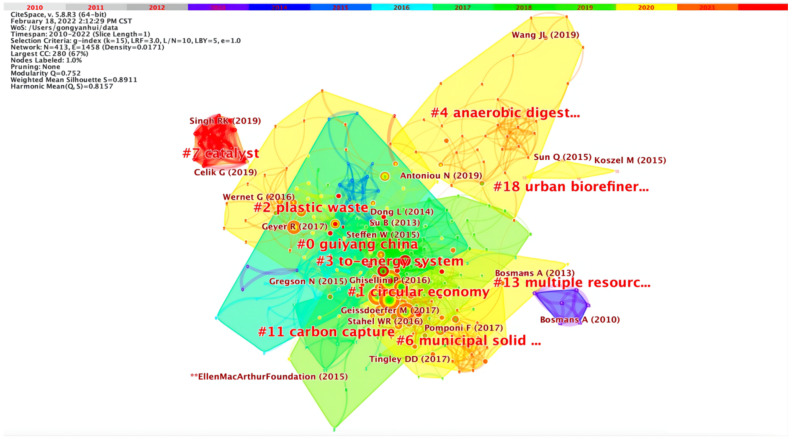
Co-citation cluster map of literature (2010–2021).

**Figure 8 ijerph-19-12508-f008:**
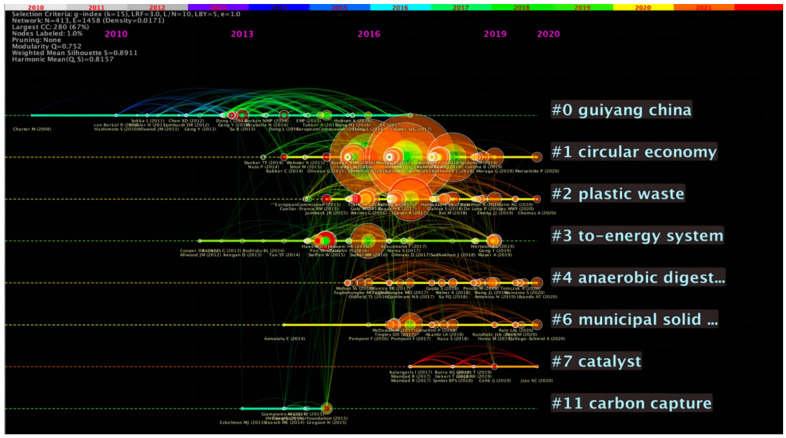
Timeline view of the co-cited articles.

**Figure 9 ijerph-19-12508-f009:**
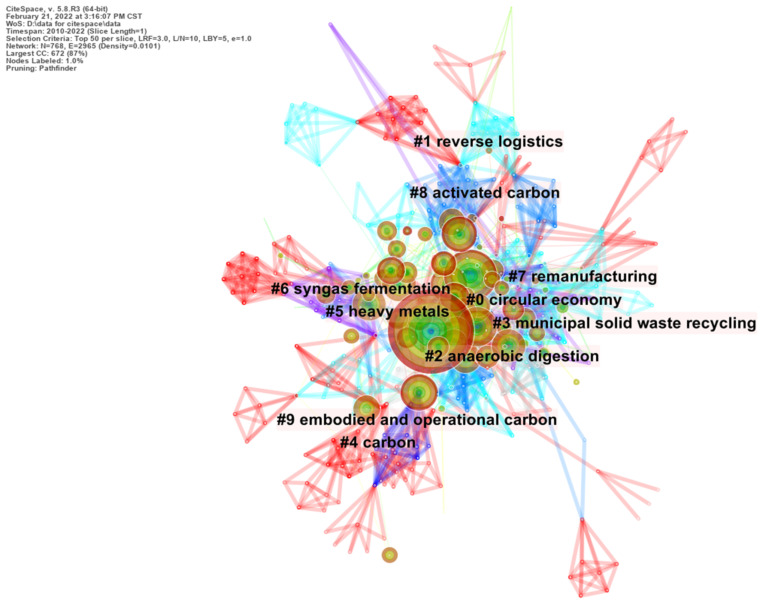
Keyword clusters of cited articles.

**Figure 10 ijerph-19-12508-f010:**
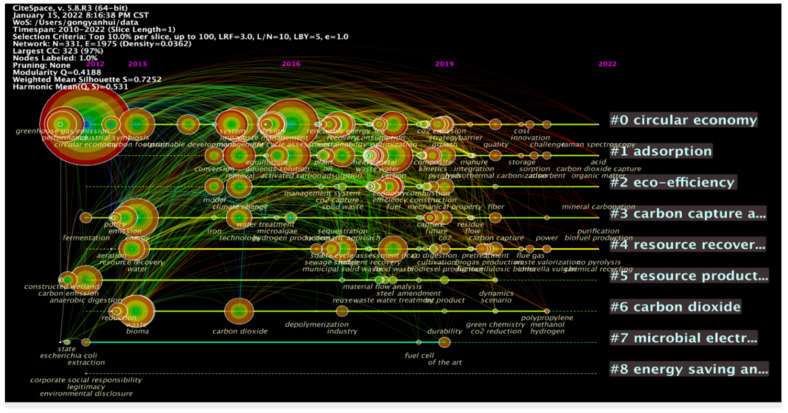
Timeline map of research hotspot.

**Table 1 ijerph-19-12508-t001:** Table of Top 10 Categories.

Wos Categories	Freq	Wos Categories	Freq
Environmental Sciences	631	Chemistry Multidisciplinary	160
Green Sustainable Science Technology	393	Environmental Studies	151
Engineering Environmental	362	Materials Science Multidisciplinary	106
Energy Fuels	231	Biotechnology Applied Microbiology	92
Engineering Chemical	171	Chemistry Physical	64

**Table 2 ijerph-19-12508-t002:** List of Top 12 Authors with the Most Published Numbers.

Author	Freq	Author	Freq
Zabaniotou A	11	Liu XX	8
Hou HY	10	Liu Z	8
Bialowiec A	9	Wang Y	8
Dong L	8	Li DD	7
Geng Y	8	Li J	7
Koziel JA	8	Purnell P	7

**Table 3 ijerph-19-12508-t003:** List of Top 10 institutions with the Most Published Numbers.

Affiliations	Freq	Affiliations	Freq
League of European Research Universities	94	Udice French Research Universities	21
Center Natinal de la Recherche Scientifique	28	Helmholtz Association	20
Consejo Superior de Investigaciones Científicas	28	Institut National de Recherche Pour l’Agriculture, l’Alimentation et l’Environnement	19
Chinese Academy of Sciences	25	Delft University of Technology	18
Consiglio Nazionale delle Ricerche	22	University of Manchester	17

**Table 4 ijerph-19-12508-t004:** List of Top 10 country or region with the Most Published Numbers.

Country/Region	Freq	Country/Region	Freq
People’s Republic of China	245	Germany	115
Italy	188	Netherlands	75
England	157	Australia	70
Spain	156	France	64
USA	128	India	64

**Table 5 ijerph-19-12508-t005:** Top 15 references with the strongest citation bursts.

References	Year	Strength	Begin	End	2010–2022
Dong L, 2014 [75], ENERG POLICY, V65, P388 DOI 10.1016/j.enpol.2013.10.019	2014	5.63	2015	2018	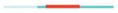
Dong L, 2013 [76], J CLEAN PROD, V59, P226DOI 10.1016/j.jclepro.2013.06.048	2013	4.63	2015	2017	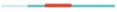
Su B, 2013 [77], J CLEAN PROD, V42, P215DOI 10.1016/j.jclepro.2012.11.020	2013	7.58	2016	2018	
Haas W, 2015 [73], J IND ECOL, V19, P765DOI 10.1111/jiec.12244	2015	4.81	2016	2020	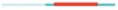
Ghisellini P, 2016 [78], J CLEAN PROD, V114, P11DOI 10.1016/j.jclepro.2015.09.007	2016	14.99	2017	2019	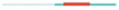
Steffen W, 2015 [79], SCIENCE, V347, P0DOI 10.1126/science.1259855	2015	7.97	2017	2020	
Pan SY, 2015 [80], J CLEAN PROD, V108, P409DOI 10.1016/j.jclepro.2015.06.124	2015	7.55	2017	2020	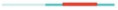
Geissdoerfer M, 2017 [81], J CLEAN PROD, V143, P757 DOI 10.1016/j.jclepro.12.048	2017	8.42	2018	2019	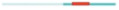
Korhonen J, 2018 [82], ECOL ECON, V143, P37DOI 10.1016/j.ecolecon.2017.06.041	2018	6.03	2018	2020	
Kirchherr J, 2017 [83], RESOUR CONSERV RECY, V127, P221DOI 10.1016/j.resconrec.2017.09.005	2017	4.82	2018	2019	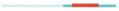
Genovese A, 2017 [84], OMEGA-INT J MANAGE S, V66, P344DOI 10.1016/j.omega.2015.05.015	2017	4.82	2018	2019	
Niero M, 2016 [85], RESOUR CONSERV RECY, V114, P18DOI 10.1016/j.resconrec.2016.06.023	2016	4.73	2018	2019	
Clark JH, 2016 [86], GREEN CHEN, V18, P3914DOI 10.1039/c6gc00501b	2016	4.51	2018	2019	
Gregson N, 2015 [87], ECON SOC, V44, P218DOI 10.1080/03085147.2015.1013353	2015	4.28	2018	2020	
Jambeck JR, 2015 [74], SCIENCE, V347, P768DOI 10.1126/science.1260352	2015	4.18	2019	2020	

**Table 6 ijerph-19-12508-t006:** Top 16 most cited papers with co-citation frequency.

Freq	Burst	Degree	Centrality	Sigma	Author	Year	Source	Vol	Page
82		24	0.04	1.00	Geissdoerfer M [81]	2017	J CLEAN PROD	143	757
73	4.81	35	0.22	2.60	Ghisellini P [78]	2016	J CLEAN PROD	114	11
65		32	0.10	1.00	Kirchherr J [83]	2017	RESOUR CONSERV RECY	127	221
48		20	0.02	1.00	Korhonen J [82]	2018	ECOL ECON	143	37
42		16	0.12	1.00	Geyer R [88]	2017	SCI ADV	3	0
34		21	0.18	1.00	Stahel WR [21]	2016	NATURE	531	435
24		21	0.05	1.00	Lieder M [89]	2016	J CLEAN PROD	115	36
23		17	0.05	1.00	Pomponi F [90]	2017	J CLEAN PROD	143	710
21		23	0.05	1.00	Bocken NMP [81]	2016	J IND PROD ENG	33	308
20		17	0.01	1.00	Kalmykova Y [91]	2018	RESOUR CONSERV RECY	135	190
19		16	0.00	1.00	Korhonen J [82]	2018	J CLEAN PROD	175	544
19		12	0.00	1.00	Murray A [92]	2017	J BUS ETHICS	140	369
19		13	0.01	1.00	Winans K [93]	2017	RENEW SUST ENERG REV	68	825
17		21	0.08	1.00	Moraga G [94]	2019	RESOUR CONSERV RECY	146	452
16	3.93	10	0.06	1.26	Steffen W [79]	2015	SCIENCE	347	0
15	3.68	9	0.02	1.07	Pan SY [80]	2015	J CLEAN PROD	108	409

**Table 7 ijerph-19-12508-t007:** Ranking of top 30 active keywords based on frequency (2010–2021).

Freq	Burst	Centrality	Time	Keyword	Freq	Burst	Centrality	Time	Keyword
596	-	0.06	2011	circular economy	56	-	0.1	2010	emission
167	-	0.02	2015	life cycle assessment	55	-	0.05	2012	greenhouse gas emission
106	-	0.07	2011	carbon	55	-	0.09	2016	food waste
105	-	0.15	2012	performance	52	2.97	0.04	2015	technology
103	-	0.08	2013	energy	52	-	0.02	2015	removal
90	-	0.03	2015	management	48	-	0.11	2017	adsorption
86	-	0.18	2012	anaerobic digestion	47	-	0.01	2016	waste management
85	2.66	0.06	2013	bioma	46	-	0.11	2015	environmental impact
81	-	0.09	2013	waste	46	-	0.03	2011	climate change
74	-	0.05	2015	activated carbon	45	-	0.02	2016	design
72	-	0.08	2014	carbon dioxide	45	-	0.04	2016	recovery
72	-	0.03	2013	carbon footprint	45	-	0.03	2016	sustainability
66	-	0.02	2014	system	43	-	0.03	2015	optimization
65	-	0.02	2015	impact	42	-	0.03	2015	waste water
57	-	0.1	2016	municipal solid waste	42	-	0.05	2015	model

## Data Availability

Not applicable.

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
