# Peer review of "Knowledge Mapping and Institutional Prospects on Circular Carbon Economy Based on Scientometric Analysis"

_ijerph, 2022, doi:10.3390/ijerph191912508_

Round 1
Reviewer 1 Report
The authors use standard bibliometric methods to mapping scientific literature on carbon economy. The CiteSpace software is used for a more advances analysis (organising papers into clusters).
The paper has several weaknesses:
The paper states that analysis of 1452 academic papers on the circular carbon economy and their references 13 over the period 1990-2021 is provided. In fact, all analyses refer to period 2010-2021 Please explain and discuss.
Some important elements are missing in the analysis
The analysis is based on the ‘Keywords’. This is OK, but the type of keywords is not explained (authors keywords, keywords assigned by the WOS, Scopus). Please explain and discuss your strategy more in detail.
The algorithm generates clusters base on keywords. Some keywords may have multiple meanings and uses (‘economy’, ‘circular’, carbon’). Nonsensical associations and/or irrelevant clusters may emerge (see: Chen, Chaomei. 2017. Science mapping: A systematic review of the literature. Journal of Data and Information Science 2: 1–40). The silhouette statistics are important for defining clusters and separate self the-standing clusters from overlapping ones. The statistics never are mentioned and discussed
Summary network statistics (upper left corner) on Figures 3 - 7suggest low-density networks. No comments are provided on the matter.
Table 4 and Figure 5 state the most important co-authorship clusters of co-operation. This is quite interesting. Please provide for detailed discussion of clusters. Why the clusters are organised in this way?
In standard papers limitations of the research are stated and their importance for research results is discussed. The paper states no limitations. Is the research so perfect?
The authors are encouraged to discuss recent uses of the CiteSpace software and comments on strengths and weaknesses of the scientometric software.
The paper badly needs thorough editing and proofreading.
Author Response
Dear reviewer:
Thank you for yourcomments concerning our manuscript entitled “Knowledge Mapping and Institutional Prospects on Circular Carbon Economy based on Scientometric Analysis” (Manuscript ID: ijerph-1903583). Those comments are all valuable and very helpful for revising and improving our paper. We have carefully considered all comments and revised our manuscript accordingly. We make some changes which we hope meet with approval. The point-by-point response is presented following this letter.

Reviewer 2 Report
Journal: International Journal of Environmental Research and Public Health
Manuscript ID: ijerph-1903583
Title: Knowledge Mapping and Institutional Prospects on Circular Carbon Economy based on Scientometric Analysis
The manuscript is well presented. However, the novelty and research gap are not well identified in the work. The fundamental problem to be solved with regard to the Circular carbon economy and the various progress made in various sectors is not clearly identified in the introduction section.
- The various circular carbon economy approach for various industries from the literature may be highlighted.
- The difference between Section 3.6 and 4 has to be clearly mentioned. A brief comparison of various literature and the research direction may be provided.
- What are the pros, cons and features of Circular carbon economy over conventional linear economy
- Relevant keywords and the parameters listed in Table 7 needs more discussion
- Research gap should be delivered on more clear way with directed necessity for the conducted research work on Circular carbon economy. Some of the references cited in the introduction are outdated. It is suggested to add 2020-22 references. The novelty and research gap of the study are not clearly identified in the work. The fundamental problem to be solved is not clearly identified.
- Please improve the quality of the Figure 8 and 10. The research hotspot and various keywords related to the circular carbon economy has to be presented well with more interesting analysis and comparison
- A good research work should contain a clear methodology and explanation of the key terms. Please improve the Methods section. The empirical and Literature review survey on CCE is very shallow. Please improve these sections
- Please provide the list of future research direction, environmental policies and legal aspects of the Circular carbon economy
- A flowchart may be included to present the whole methodology if the work.
- What are the “Current challenges, recent advances and applications of Circular carbon economy
- The conclusion section is weak with little or no numbers to support the findings. Please include the limitations, future scope and practical implication of the study
Author Response
Dear reviewer:
Thank you for your comments concerning our manuscript entitled “Knowledge Mapping and Institutional Prospects on Circular Carbon Economy based on Scientometric Analysis” (Manuscript ID: ijerph-1903583). Those comments are all valuable and very helpful for revising and improving our paper. We have carefully considered all comments and revised our manuscript accordingly. We make some changes which we hope meet with approval. The point-by-point response is presented following this letter.

Reviewer 3 Report
This is a well written and well structured paper, which presents significant and credible findings. In their conclusion the author states: "We concluded that the research on circular carbon economy experienced a relatively slow growth from 2010 to 2016, and an explosive growth from 2018 to 2021 in the aspect of the overall research trend. The research direction also shifted from the construction of macroknowledge structure to the analysis of micro-disciplinary content, and the characteristics 563 of interdisciplinary and refined research became increasingly prominent. The number of authors' articles and their cooperation reflected that the core group of authors of circular carbon economy research had not yet formed, and there was less cooperation among authors from various countries, while more cooperation among Chinese scholars. The participation of social institutions on circular carbon economy is low, and universities holds 568 the mainstream positions of academic research on circular carbon economy."
I think it would be of interest to the reader if the authors tried to explain these findings. For example, which factors could have led to the slow growth of the literature and the existing patterns of scientific work. Does this map for instance to the Kuhnian ideas of a scientific revolution or paradigm shift, or could other frameworks (e.g. punctuated equilibrium theory explain this)
Author Response

(The authors gave the same response as above.)

Round 2
Reviewer 1 Report
The authors addressed all comments. Quality of the manuscript improved.
Authors should address some remaining minor issues, e.g. ‘Secondly, this study takes English as the main language, only analyzes English journals,...’ (p. 22). They probably mean journals publishing in English (and not journals published in England).
The authors may consider editing language by native English speaker.
I recommend this paper for publication
Author Response
Thank you very much for your professional advice. We have further revised the manuscript according to your comments. We sincerely hope that it can meet the standards of journals.
Response to Reviewer 1 Comments
Point 1: The authors addressed all comments. Quality of the manuscript improved.
Authors should address some remaining minor issues, e.g. ‘Secondly, this study takes English as the main language, only analyzes English journals,...’ (p. 22). They probably mean journals publishing in English (and not journals published in England).
The authors may consider editing language by native English speaker.
I recommend this paper for publication
Response 1: We gratefully appreciate for your valuable suggestion. First of all, we have modified the ambiguous sentence you pointed out based on your helpful comments. (see details, line 739) Then, the manuscript has also been thoroughly revised, and we really hope it has been substantially improved. (see details, line 18, 24, 62, 99, 109, 120, 128, 139, 179, 180, 195, 211, 241, 286, 301, 307, 314, 330, 344, 353, 358, 370, 378, 388, 392, 439, 448, 456, 473, 505, 508, 560, 615, 634, 659, 675, 745, 826) We would like to thank you again for taking the time to review our manuscript.
